# Nutritional composition and antioxidant properties of the fruit of *Berberis heteropoda* Schrenk

**Jixiang Sun**[1,2], **Qian Li**[2], **Jianguang Li**[1]*, **Jing Liu**[2], **Fang Xu**[1]

**1** College of Pharmacy, Xinjiang Medical University, Urumqi, China, **2** People's Hospital of Xinjiang Uygur Autonomous Region, Urumqi, China

* xjykdx_ljg@163.com

## Abstract

### Objective

This study assessed the major nutrients and antioxidant properties of *Berberis heteropoda* Schrenk fruits collected from the Nanshan Mountain area of Urumqi City, Xinjiang Uygur Autonomous Region, China.

### Methods and materials

We assessed the basic nutrients, including amino acids, minerals, and fatty acids, and determined the total phenol, flavonoid, and anthocyanin contents of the extracts.

### Results

The analytical results revealed the average water (75.22 g/100 g), total fat (0.506 g/100 g), total protein (2.55 g/100 g), ash (1.31 g/100 g), and carbohydrate (17.72 g/100 g) contents in fresh *B. heteropoda* fruit, with total phenol, flavonoid, and anthocyanin contents of *B. heteropoda* fruits at 68.55 mg gallic acid equivalents/g, 108.42 mg quercetin equivalents/g, and 19.83 mg cyanidin-3-glucoside equivalent/g, respectively. Additionally, UPLC-Q-TOF-MS$^E$ analysis of polyphenols in *B. heteropoda* fruit revealed 32 compounds.

### Conclusion

*B. heteropoda* fruits may have potential nutraceutical value and represent a potential source of nutrition and antioxidant phytochemicals in the human diet.

## Introduction

*Berberis heteropoda* Schrenk is a shrub of the family Berberidaceae, which is distributed in the Altai, Tianshan, and Baluke mountains of the Xinjiang Uygur Autonomous Region, China, as well as in Mongolia and Kazakhstan [1]. The roots, bark, stems, and fruits of *B. heteropoda* are traditionally used as an herbal medicine, and the fruits in particular have historically been

**Data Availability Statement:** All relevant data are within the paper and its Supporting information files.

**Funding:** This research was funded by the Natural Science Foundation of Xinjiang Uygur Autonomous Region, grant number 2021D01C177.

**Competing interests:** The authors have declared that no competing interests exist.

consumed as a tea [2, 3]. In modern times, studies have confirmed that this fruit can be used to treat dysentery, enteritis, pharyngitis, stomatitis, eczema, and hypertension [4, 5]. Because the nutritional and antioxidant properties of *B. heteropoda* fruit are related to its molecular and secondary metabolite content, there are potential benefits to its consumption.

In addition to proteins, fats, dietary fiber, minerals, and other nutrients, plants also contain numerous phenolic components that can play an important role in human health [6]. Polyphenols are secondary metabolites produced by plants and often observed in vegetables, fruits, and forages [7]. Phenolic compounds are effective at preventing oxidation at the cellular and physiological levels, with their antioxidant capacity determined based on the arrangement of hydroxyl and carbonyl groups in their structures, as well as the gain and loss of electrons from hydrogen atoms to reduce free radicals and form stable phenoxy groups [8, 9]. Flavonoids are major components of plant polyphenols and play important roles in antioxidant effects, including in reduction reactions as a hydrogen donor for singlet oxygen quenching and metal chelation. Thus, evaluating the polyphenol, flavonoid, and antioxidant contents of *B. heteropoda* Schrenk fruit and evaluating its medicinal and nutritional value are important.

A previous study focused on the anthocyanin composition of *B. heteropoda* fruit [3]; however, the nutritional and phenolic composition of *B. heteropoda* remains unclear. Therefore, in the present study, we assessed the major nutrient content and antioxidant properties of *B. heteropoda* and investigated the active components of the plant, as well as how this information can guide its nutritional use.

## Materials and methods

### Plant material

A total of 3 kg of ripe *B. heteropoda* fruit was collected from dozens of shrubs in a ravine in the Nanshan Mountain area of Urumqi City, Xinjiang Uygur Autonomous Region, China (latitude 89°29′36″E, longitude 43°27′32″N), in September 2019. The specimens were identified by expert Lude Xin from Xinjiang Medical University, and a voucher specimen (WR2101079001) was deposited in the Institute of Clinical Nutrition, People's Hospital of Xinjiang Autonomous Region. The fruits were then transported to the laboratory for a pre-cooling treatment (−20 °C) 2 h after harvest. The fruits for study were selected after being combed, and we ensured that all of the selected fruits were even and full, with uniform size and maturity. Fruits without mechanical damage, rot, or other miscellaneous defects were selected for further analysis. Subsequently, the stem and seeds were removed, and fruits were placed in dark storage at −20 °C until further use.

### Standards and reagents

The reagents 1,1-diphenyl-2-picryl-hydrazl (DPPH) and 2,2-azinobis-(3-ethylbenzthiazoline-6-sulfonic acid) (ABTS) were purchased from Shanghai Macklin Biochemical Co., Ltd. (Shanghai, China). Gallic acid and rutin standards were purchased from Chengdu Munster Biotechnology Co., Ltd. (Chengdu, China). Folin–Ciocalteu's phenol reagent was purchased from Tianjin Kaitong Chemical Reagent Co., Ltd. (Tianjin, China). Anhydrous methanol, anhydrous ethanol, concentrated hydrochloric acid, sodium nitrite, sodium hydroxide, sodium carbonate, and ferrous sulfate were obtained from Sinopharm Chemical Reagent Co., Ltd. (Shenyang, China).

### Nutritional composition

**Determination of general nutrients.** Crude protein content was determined using the Kjeldahl method according to Chinese National Standard (CNS) GB/T5009.5–2016

"Determination of protein in food." Ash content was measured using the muffle furnace burning method according to CNS GB 5009.4–2016 "Determination of ash in food." Crude fat was determined using the Soxhlet extraction method according to CNS GB 5009.6–20163 "Determination of fat in food." Moisture was measured using the direct drying method according to CNS GB5009.3–2016 "Determination of moisture in food." Carbohydrate content was determined based on CNS NY/T 2332–2013. The total energy of each sample was calculated as follows: Total Energy (kJ) = 17 × (g crude protein + g total carbohydrate) + 37 (g crude fat) [10].

**Mineral composition.**   Mineral and element contents were determined according to CNS GB5009.268–2016 "Determination of multi-elements in food" using inductively-coupled plasma (ICP)-mass spectrometry [MS; 5110 ICP optical emission spectrometer (OES); Agilent Technologies, Santa Clara, CA, USA]. Briefly, a 1.0-g slurry sample was digested in 2 mL of concentrated $HNO_3$ in a microwave oven and then diluted with distilled water to 25 mL. The solution was filtered before storage, and a blank digest was performed in a similar manner. The blank solution and the test solution were each injected into the ICP OES to determine the contents of K, Ca, Na, Mg, Fe, Cu, Zn, and P.

**Amino acid analysis.**   Amino acid contents were measured by an automatic amino acid analyzer (L-8900; Hitachi, Tokyo, Japan) according to CNS GB 5009.124–2016 "Determination of amino acid in food." Continuous flash evaporation at reduced pressure was used to remove excess acid, and the sample was dissolved in citrate buffer (pH 2.2) [11].

**Fatty acids.**   Fatty acid composition and content was determined by gas chromatography–MS (7890B/7000D; Agilent Technologies) according to CNS GB 5009.168–2016. Triglyceride undecarbonate was used as an internal reference, 37 different fatty acid methylester standard solutions were used as external references. The fatty acid content was quantitatively measured using chromatographic peaks.

## Extraction and quantification of Total Phenol Content (TPC), Total Flavonoid Content (TFC), and Total Anthocyanin Content (TAC)

**Extraction.**   Sample extraction was performed using a previously reported method [12], with slight modification. Briefly, 1.0 g of *B. heteropoda* fruit was added to 30 mL of 70% acidified ethanol (0.1% HCl, v/v), and the solution was extracted three times under ultrasonic conditions (40 kHz, 100 W) for 30 min at 25 ˚C. The mixture was then centrifuged at 1000 r/min for 15 min, and the supernatant was collected. The residue was subsequently extracted twice, all of the collected supernatant was mixed together and concentrated under vacuum, and the extraction was preserved at −20 ˚C until further analysis. The solvents used for fruit extraction included methanol, acetone, and ethanol. The final extract was used for the quantification of TPC, TFC, TAC, and antioxidant activity.

**TPC Determination.**   The TPC was measured using Folin–Ciocalteu's phenol reagent with the colorimetric method [13]. Briefly, 0.5 mL of reagent and 1.5 mL of sodium carbonate solution (10%, w/v) were added to 1 mL of *B. heteropoda* fruit extract, followed by immediate addition of 8 mL of distilled water and incubation for 10 min in a water bath at 75 ˚C. The absorbance was the measured using an ultraviolet–visible (UV–vis) spectrophotometer (New Century T6; Persee Analytics, Beijing, China) at 760 nm. We generated a standard curve of the absorbance value of gallic acid solution, and then TPC was determined as milligram of gallic acid equivalent per gram of fresh fruit mass.

**TFC Determination.**   The TFC was measured using rutin as a reference standard with the aluminum nitrate method [14]. Briefly, 0.5 mL of *B. heteropoda* fruit extract was added to 1 mL of sodium nitrite and incubated for 6 min, followed by mixture with 1 mL of 10% aluminum nitrate and then incubation for another 6 min. We then added 10 mL of 1.0 M sodium

hydroxide, adjusted the volume of water to 20 mL, and incubated the solution for 15 min. UV–vis spectrometry was then used to detect the absorbance at 510 nm and generate a standard curve. TFC was denoted as milligram of rutin equivalent per gram of weight of fresh fruit mass.

**TAC Determination.** The TAC was determined by the pH differential method [15]. Briefly, 2 mL of fruit extract was added to a centrifuge tube for centrifugation at 1000 r/min for 5 min, after which 0.5 mL of supernatant was added into two 10-mL volumetric flasks: one with a buffer at pH 1.0 and the other with a buffer at pH 4.5. The absorbance at 517 nm and 700 nm was measured after a 15-min incubation, and data were expressed as milligram of cyanidin-3-glycoside equivalents per gram of fresh fruit mass. The TAC was calculated according to the following formula:

$$A = [(A_{517} - A_{700})_{pH\,1.0} - (A_{517} - A_{700})_{pH\,4.5}], BHSTAC(mg/g) = A \times MW \times DF \times \frac{1}{\varepsilon \times L} \times \frac{V}{M},$$

where MW represents the molecular weight of centrinin-3-glycoside [449.2 g/mol; according to the centrothrin-3-glycoside molar extinction coefficient (26900 $L \cdot cm^{-1} \cdot mol^{-1}$)], DF represents diluted multiples, L denotes absorption pool thickness (1 cm), V represents extraction volume (mL), and M denotes the weight of peel powder.

## Measurements of antioxidant capacity

**DPPH free radical assay.** The DPPH free radical-scavenging assay was performed according to the method described by Vlase et al. [16]. Briefly, *B. heteropoda* fruit extract was dissolved in 70% ethanol at different concentrations and mixed with 2 mL of a freshly prepared ethanol solution of DPPH free radicals (100 μM). The solution was mixed vigorously and stored in darkness at room temperature for 30 min, followed by UV–vis spectrometry detection of the absorbance at 517 nm. The positive control group was measured using vitamin C ($V_C$). The results were expressed as half maximal inhibitory concentration ($IC_{50}$), which was used to indicate the corresponding concentration of the extract when the anti-oxidation free radical-scavenging capacity was 50%:

$$\text{DPPH free radical scavenging rate} = \left(1 - \frac{A_S - A_0}{A_C}\right) \times 100\%,$$

where $A_C$ denotes the absorbance value of the control, $A_0$ represents the absorbance value of the blank, and $A_S$ is the absorbance value of the sample.

**ABTS free radical assay.** The ABTS free radical-scavenging assay was performed according to the method described by Lyu et al. [17]. Briefly, 2 mL of 10 mM potassium persulfate solution and 2 mL of 10 mM ABTS free radical solution were mixed and then stored in the dark for 12 h. Ethanol was then added to the mixed solution until its UV–vis absorbance value reached 0.700 ± 0.020 at 736 nm. Subsequently, 2 mL of *B. heteropoda* fruit extract or ascorbic acid solution was mixed vigorously with 2 mL of ABTS working solution and stored in the dark at room temperature for 10 min. The $IC_{50}$ values of the sample extract were calculated based on the concentration and capacity designated by the free radical-scavenging curves:

$$\text{ABTS free radical scavenging rate} = \frac{A_C - A_S}{A_C} \times 100\%$$

where $A_C$ represents the absorbance value of the control, and $A_S$ denotes the absorbance value of the sample.

**Hydroxyl free radical assay.** The hydroxyl free radical assay was performed according to the method described by Liang et al. [18]. Briefly, 0.5 mL of 7.5 mM ferrous sulfate heptahydrate, 0.5 mL of 7.5 mM salicylic acid, 1 mL of *B. heteropoda* fruit extract, and 0.2 mL of 30% hydrogen peroxide were mixed and incubated for 30 min in a water bath at 37 ˚C. After cooling, the absorbances of the hydroxyl radical sample, blank, and control groups were determined at 510 nm on the UV–vis spectrometer, and hydroxyl radical scavenging activity (HRSA) was determine as follows:

$$\text{HRSA}(\%) = \left(\frac{A_S - A_C}{A_0 - A_C}\right) \times 100\%$$

where $A_C$ denotes the absorbance value of the control, $A_0$ represents the absorbance value of the blank, and $A_S$ represents the absorbance value of the sample.

**Superoxide anion free radical assay.** The superoxide anion free radical assay was performed according to the method described by Liu et al. [15]. Briefly, 4.5 mL of 50 mM Tris–hydrochloric acid and 1 mL of *B. heteropoda* fruit extract were mixed and incubated 15 min in a water bath at 25 ˚C, followed by the addition of 0.4 mL of 5 mM pyrogallic acid and incubation for 5 min in a water bath at 25 ˚C. Subsequently, 0.1 mL of 8 M hydrochloric acid was added to terminate the reaction, and the absorbance values of the sample, blank, and control were measured at 325 nm on the UV–vis spectrometer to determine the following rate:

$$\text{Superoxide anion scavenging rate} = \frac{A_C - A_S}{A_S} \times 100\%$$

where $A_C$ denotes the absorbance value of the control, and $A_S$ represents the absorbance value of the sample.

## Chromatography and mass spectrometry

**Chromatographic conditions.** Chromatographic separations were performed using an ultra-high performance liquid chromatography (UPLC) 1290 system with a Waters UPLC BEH C18 column (1.7 μm 2.1 × 100 mm; Agilent Technologies). The flow rate was set to 0.4 mL/min, and the sample-injection volume was set to 5 μL. The mobile phases comprised 0.1% formic acid in water (A) and 0.1% formic acid in acetonitrile (B). The multi-step linear elution gradient program was as follows: 0 to 3.5 min, 95% to 85% A; 3.5 to 6 min, 85% to 70% A; 6 to 6.5 min, 70% to 70% A; 6.5 to 12 min, 70% to 30% A; 12 to 12.5 min, 30% to 30% A; 12.5 to 18 min, 30% to 0% A; 18 to 25 min, 0% to 0% A; 25 to 26 min, 0% to 95% A; and 26 to 30 min, 95% A.

**MS conditions.** We used a Q Exactive Focus mass spectrometer coupled with Xcalibur software (Thermo Fisher Scientific, Waltham, MA, USA) was employed to obtain MS and MS/MS data in independent data acquisition mode. During each acquisition cycle, the mass range was set to a range of 100 to 1500, the top three data points in every cycle were screened, and the corresponding MS/MS data were further acquired. The following parameters were used: sheath gas-flow rate, 45 Arb; auxiliary gas-flow rate, 15 Arb; capillary temperature, 400 ˚C, full MS resolution, 70,000; MS/MS resolution, 17,500; collision energy, 15/30/45 in normalized collision energy mode; and spray voltage, 4.0 kV (positive) or −3.6 kV (negative).

## Statistical analysis

All experimental data were collected in triplicate, and data were expressed as the mean ± standard deviation. Statistical analyses were performed using GraphPad Prism (v.7.0; GraphPad Software, La Jolla, CA, USA) and SPSS (v.23.0; IBM Corp. Armonk, NY, USA).

**Table 1. Proximate nutritional composition of fresh *Berberis heteropoda* fruit.**

| Composition (Unit) | |
|---|---|
| Water (g/100 g) | 75.22±1.75 |
| Total fat (g/100 g) | 0.506±0.02 |
| Total Protein (g/100 g) | 2.55±0.03 |
| Ash (g/100 g) | 1.31±0.04 |
| Total sugars (g/100 g) | 0.05±0.00 |
| Carbohydrates (g/100 g) | 17.72±0.52 |
| Total Energy (kJ) | 363.52±7.51 |

## Results

### Nutritional composition of *B. heteropoda* fruit

**Proximate composition of *B. heteropoda* fruit.** The major nutrients of *B. heteropoda* fruit are summarized in Table 1 and S1 File. The major components were identified as water, crude fiber, and total protein, with values of 75.22±1.75 g/100 g, 17.72±0.52 g/100 g, and 2.55 ±0.03 g/100 g, respectively Ash content was 1.31±0.04 g/100 g, indicating that the fruit is rich in minerals. The total sugar and total fat contents were 0.05±0.00 g/100 g and 0.51±0.02 g/100 g, respectively, and the energy content per 100 g of fruit was 363.52 kJ.

**Minerals.** We detected a total of eight minerals in *B. heteropoda* fruit (Table 2 and S1 File). We found that K (582.67±8.02 mg/100 g) was the most abundant element [19], with Ca (78.5±1.62 mg/100 g), P (73.24±1.72 mg/100 g), and Mg (30.61±0.56 mg/100 g) also abundant.

**Amino acids.** The 16 amino acids identified in *B. heteropoda* fruit are shown in Table 2 and S1 File. Glutamic acid was the most abundant amino acid, followed by aspartic acid, arginine, lysine, and glycine. The fruit contained six types of essential amino acids (EAAs) to a value of 0.9 g/100 g fruit weight and accounting for 31.8% of the total amino acids, with the content of the remaining 10 non-EAAs (NEAAs) at 1.93 g/100 g fruit weight.

**Fatty acids.** The fatty acid content in *B. heteropoda* fruit is presented in Table 3 and S1 File. We identified a total of 10 different fatty acids, including saturated and unsaturated varieties. Tetrahexanoic acid (C24:0) was the dominant fatty acid, followed by octadecentrienoic acid (C18:3) and octadecadienoic acid (C18:2). The unsaturated fatty acid (UFA) content was slightly higher than that of the saturated fatty acid (SFA) content (51.51% vs. 48.48%).

### TPC, TFC, and TAC

The TPC, TFC, and TAC values for *B. heteropoda* fruit are shown in Fig 1 and S2 File. The methods used to determine flavonoid and polyphenol contents showed a good linear relationship within the measurement range ($r^2$ = 0.995 and 0.999, respectively), with the following regression equations used: y = 0.0109x + 0.0157 and y = 0.067x − 0.0173, respectively. The extraction effect of each solvent (high to low) was methanol > acetone > ethanol for total phenol, total flavonoids, and total anthocyanins. Using methanol as the extraction solvent yielded TFC, TPC, and TAC values of 108.42 mg/g, 68.55 mg/g, and 19.83 mg/g fresh fruit weight, respectively. These results suggested that methanol as the extraction solvent obtained higher total flavonoid and total phenol values.

### Antioxidant activity of *B. heteropoda* fruit extract

The antioxidant activity of the *B. heteropoda* fruit extracts was evaluated using $V_C$ as the control, with the $IC_{50}$ values for the DPPH free radical-, ABTS radical-, •OH-, $O_2$•-scavenging

**Table 2. Nutritional composition (minerals and amino acids) of *Berberis heteropoda* fruit.**

| Minerals | Composition (mg/100 g FW) | Total minerals (%) |
|---|---|---|
| Na | 1.38±0.03 | 0.18 |
| K | 582.67±8.02 | 75.73 |
| Ca | 78.5±1.62 | 10.20 |
| Cu | 0.27±0.01 | 0.04 |
| Zn | 0.59±0.01 | 0.08 |
| Fe | 2.31±0.05 | 0.30 |
| Mg | 30.61±0.56 | 3.97 |
| P | 73.24±1.72 | 9.51 |
| Total minerals | 769.843 | |
| **Amino acids** | **Composition (g/100 g FW)** | **Total amino acids (%)** |
| Phenylalanine | 0.12±0.01 | 4.24 |
| Alanine | 0.17±0.01 | 6.01 |
| Methionine | 0.015±0.00 | 0.53 |
| Proline | 0.2±0.02 | 7.07 |
| Glycine | 0.21±0.01 | 7.43 |
| Glutamic acid | 0.53±0.01 | 18.74 |
| Arginine | 0.22±0.01 | 7.78 |
| Lysine | 0.21±0.00 | 7.43 |
| Tyrosine | 0.11±0.00 | 3.89 |
| Leucine | 0.18±0.01 | 6.36 |
| Serine | 0.13±0.01 | 4.60 |
| Threonine | 0.13±0.01 | 4.60 |
| Aspartic acid | 0.27±0.01 | 9.55 |
| Valine | 0.15±0.01 | 5.30 |
| Histidine | 0.073±0.00 | 2.58 |
| Isoleucine | 0.11±0.01 | 3.89 |
| Total amino acids | 2.828 | |

FW, fruit weight.

**Table 3. Fatty acid content in *Berberis heteropoda* fruit.**

| Fatty acids | Formula | Composition (g/100 g fatty acid) | Proportion (%) |
|---|---|---|---|
| Myristic acid (C14:0) | $C_{14}H_{28}O_2$ | 0.0039 | 1.41 |
| 2-methyl-heptanoic acid (C8:0) | $C_8H_{16}O_2$ | 0.0019 | 0.71 |
| Hexadecanoic acid (C16:0) | $C_{16}H_{32}O_2$ | 0.0285 | 10.35 |
| Stearyl acid (C18:0) | $C_{18}H_{36}O_2$ | 0.0041 | 1.50 |
| Octadecenoic acid (C18:1)* | $C_{18}H_{32}O_2$ | 0.0263 | 9.55 |
| Octadecadienoic acid (C18:2)* | $C_{18}H_{32}O_2$ | 0.0526 | 19.11 |
| Octadecentrienoic acid (C18:3)* | $C_{18}H_{30}O_2$ | 0.0630 | 22.86 |
| Arachidic acid (C20:0) | $C_{20}H_{40}O_2$ | 0.0038 | 1.38 |
| Docosanoic acid (C22:0) | $CH_3(CH2)_{20}COOH$ | 0.0112 | 4.06 |
| Tetrahexanoic acid (C24:0) | $CH_3(CH2)_{22}COOH$ | 0.0801 | 29.08 |
| Subtotal | | 0.2754 | 100.00 |

* UFAs

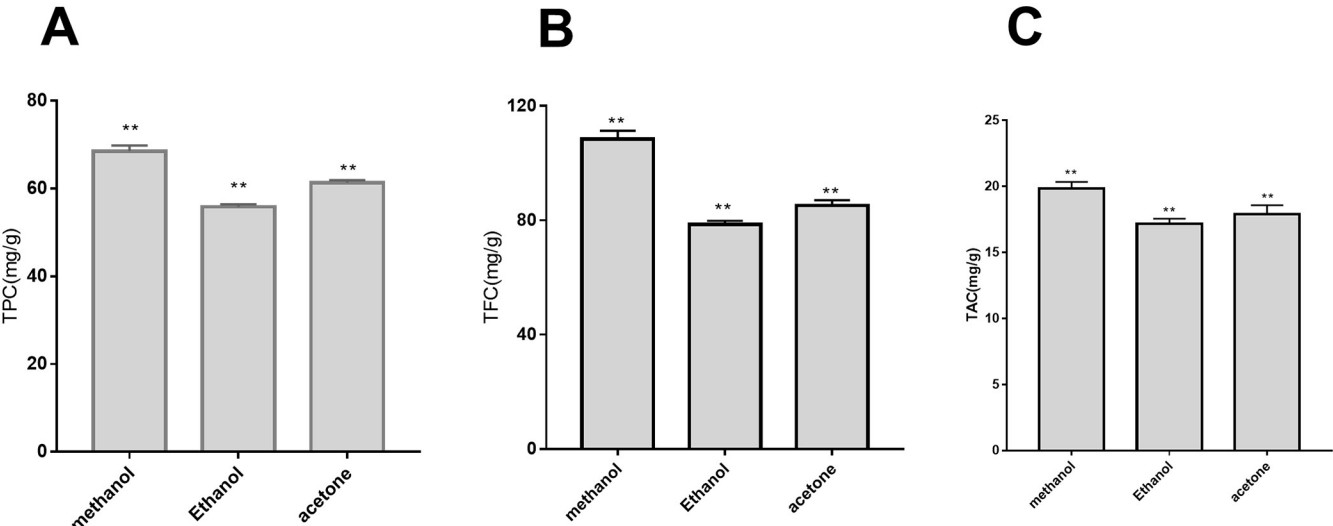

**Fig 1. Quantification of TPC (A), TFC (B), and TAC (C) in *Berberis heteropoda* shrub extract.** Comparison of the extraction effect of methanol, acetone, and ethanol. **P < 0.05.

abilities at 20.27±0.26 μg/mL, 13.89±0.13 mg/mL, 5.81±0.13 mg/mL, and 0.57±0.02 mg/mL, respectively (Fig 2 and S2 File). We observed that methanol extract had the best antioxidant activity, with $IC_{50}$ values for DPPH radical-, hydroxyl radical-, ABTS radical-, and superoxide anion radical-scavenging abilities at 20.13 μg/mL, 5.44 mg/mL, 8.79 μg/mL, and 1.35 mg/ mL, respectively. The $IC_{50}$ values for methanol extraction were higher than those of $V_C$ but lower than those of ethanol and acetone extraction. The ranking from high to low according to free radical-scavenging ability was methanol > acetone > ethanol and suggested that *B. heteropoda* fruit extracts showed good antioxidant activity based on effective free radical scavenging.

### Identification of phenols in *B. heteropoda* fruit extract using chromatography and MS

The UPLC-quadrupole time-of-flight (Q-TOF)-MS spectra indicated that the compounds in the extract of *B. heteropoda* fruit were primarily identified within 2 min to 10 min and when the mobile phase was at 15% to 70% ethyl alcohol solution, indicating that the polyphenols of *B. heteropoda* fruit belonged to polar compounds (Table 4, S3 File and S1 Fig) [10, 12, 20–38].

### Discussion

A previous study on the anthocyanin composition of *B. heteropoda* fruit considered it as a potential anthocyanin pigment source [3] and focused on chemical characterization of *B. heteropoda* fruit; however, there has been a comprehensive investigation of the overall nutritional composition of the fruit. The present study systematically evaluated the major nutrients and antioxidant properties of *B. heteropoda* fruits and found them to be rich in various nutrients, thereby providing evidence for their potential health-related or nutritional use. Moreover, we identified a total of 32 polyphenols in *B. heteropoda* fruit extract.

The results revealed that *B. heteropoda* fruit exhibits nutritional properties suggesting potential nutraceutical value. The major nutrients of *B. heteropoda* fruit were comparable to those of wolfberry (*Lycium ruthenicum* Murr.), which is a wild plant and widely observed in Xinjiang [39]. Additionally, the protein content of *B. heteropoda* fruit was higher than that in

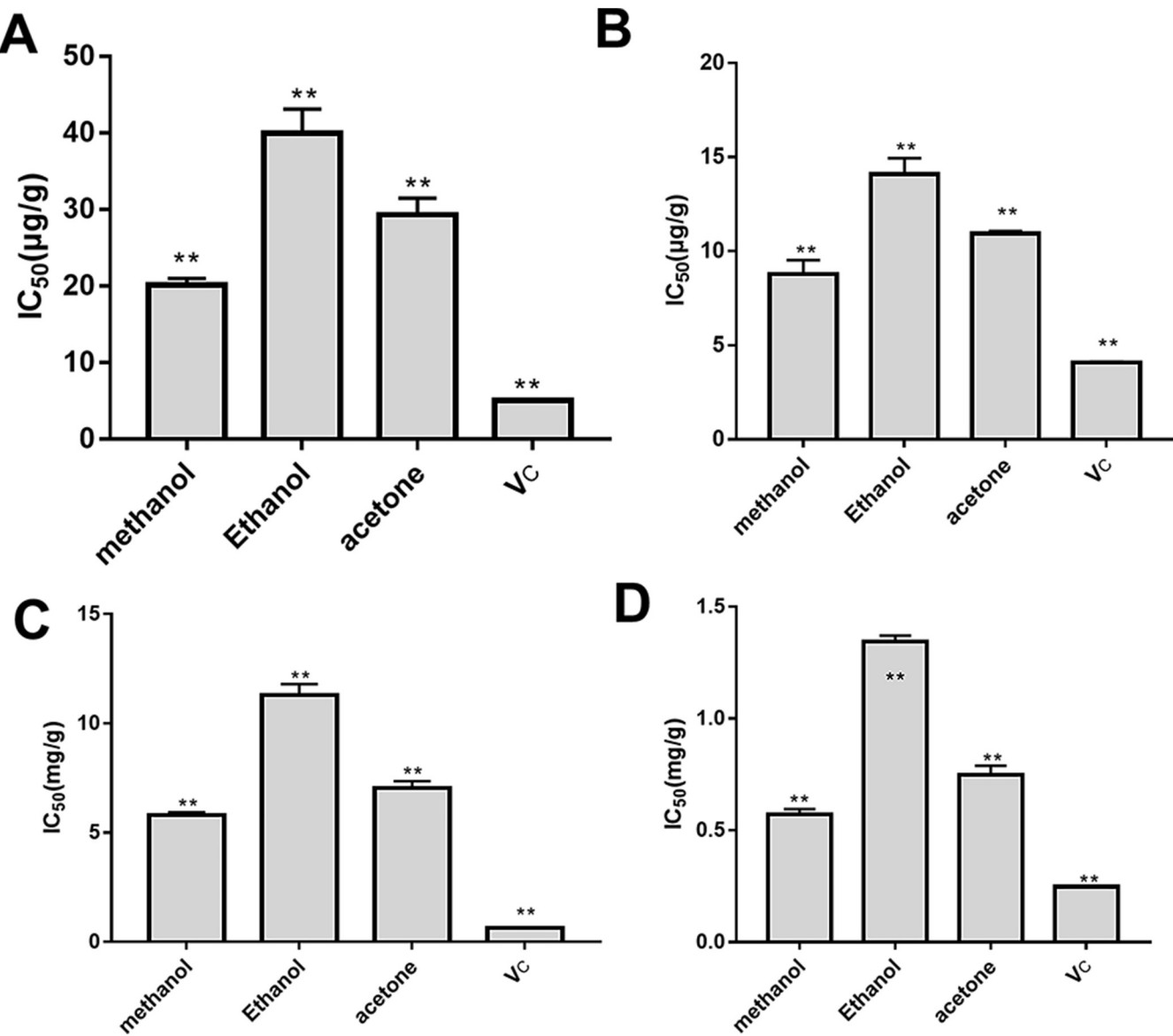

**Fig 2. IC$_{50}$ values (mg/mL) of different extracts on free radicals.** Comparison of IC$_{50}$ values using methanol, ethanol, acetone, and Vc for extraction and scavenging of (A) DPPH radical, (B) ABTS radical, (C) hydroxyl radical, and (D) superoxide anion radical. $^{**}$P < 0.05.

black mulberry (1.17±0.06%) [19], and the contents of fat and sugar in *B. heteropoda* fruit were low, suggesting a low risk for causing obesity and consideration for use as a functional food or medicine rather than an edible fresh fruit due to its poor taste.

We found that *B. heteropoda* fruit contains numerous minerals, including Na, K, Ca, Cu, Zn, Fe, Mg, and P. Previous studies demonstrate that these minerals play important roles in the physiological function of human tissues, maintaining cellular osmotic pressure, supporting the pH balance of the body, and regulating specific physiological functions as cofactors [40, 41]. Additionally, we observed that the Na:K ratio in *B. heteropoda* fruit was 0.002, which could promote the prevention of hypertension [42]. These findings suggest that *B. heteropoda* fruit might be considered helpful for controlling blood pressure.

**Table 4. Characterization of phenolic compounds of *Berberis heteropoda* fruit by UPLC-Q-TOF-MS$^E$.**

| Compound | $t_R$/min | Ionization mode | Identification | Molecular formula | MS(m/z) | MS2(m/z) |
|---|---|---|---|---|---|---|
| 1 | 1.52 | [M-H]$^-$ | Corilagin | C27H22O18 | 633.0787 | 261.667; 181.051 |
| 2 | 2.96 | [M-H]$^-$ | Petunidin-3-O-beta-glucopyranoside | C22H23O12 | 477.1030 | 299.013; 314.043 |
| 3 | 3.67 | [M-H]$^-$ | Cianidanol | C15H14O6 | 289.0719 | 245.0827 |
| 4 | 4.53 | [M-H]$^-$ | Gossypetin-8-C-glucoside | C21H20O13 | 479.0835 | 316.0244; 271.216 |
| 5 | 5.03 | [M-H]$^-$ | Syringetin-3-O-glucoside | C23H24O13 | 507.1144 | 301.067; 345.0604 |
| 6 | 5.50 | [M-H]$^-$ | Myricetin-3-O-galactoside | C21H20O13 | 479.0834 | 115.0551; 133.014 |
| 7 | 5.80 | [M-H]$^-$ | kaempferol 7-O-glucoside | C21H20O11 | 447.0926 | 285.0365 |
| 8 | 5.84 | [M-H]$^-$ | Syringetin-3-O-galactoside | C23H24O13 | 507.1143 | 344.053; 273.032 |
| 9 | 5.85 | [M-H]$^-$ | Flavanomarein | C21H22O11 | 449.1095 | 287.0572; 150.0037 |
| 10 | 5.89 | [M-H]$^-$ | Luteolin | C15H10O6 | 285.0393 | 151.0022; 133.0302 |
| 11 | 5.91 | [M-H]$^-$ | Spiraeoside | C21H20O12 | 463.089 | 301.034; 179.0188 |
| 12 | 6.02 | [M-H]$^-$ | Myricetin | C15H10O8 | 317.0300 | 137.0248; 151.00568 |
| 13 | 6.14 | [M-H]$^-$ | Luteolin-4'-O-glucoside | C21H20O11 | 447.0927 | 285.0388 |
| 14 | 6.20 | [M-H]$^-$ | Dihydromyricetin | C15H12O8 | 319.0458 | 150.999; 107.0111 |
| 15 | 6.73 | [M-H]$^-$ | Morin | C15H10O7 | 301.0356 | 165.02 |
| 16 | 6.97 | [M-H]$^-$ | Quercetin | C15H10O7 | 301.0357 | 121.0272; 151.0038; 178.9974 |
| 17 | 7.97 | [M-H]$^-$ | Kaempferol | C15H10O6 | 285.0408 | |
| 18 | 8.04 | [M-H]$^-$ | Kaempferide | C16H12O6 | 299.0556 | 284.0329; 256.036 |
| 19 | 8.19 | [M-H]$^-$ | Isorhamnetin | C16H12O7 | 315.0507 | 300.029 |
| 20 | 9.56 | [M-H]$^-$ | Galangin | C15H10O5 | 269.0455 | 225.0558 |
| 21 | 4.52 | [M+H]$^+$ | Genistein | C15H10O5 | 271.0588 | 121.028 |
| 22 | 4.75 | [M+H]$^+$ | Flavokawain B | C17H16O4 | 285.1122 | 249.1829; 267.141 |
| 23 | 4.85 | [M+H]$^+$ | Epicatechin | C15H14O6 | 291.0858 | 123.0446; 139.039 |
| 24 | 5.05 | [M+H]$^+$ | Herbacetin | C15H10O7 | 303.0478 | 257.042 |
| 25 | 5.08 | [M+H]$^+$ | Dihydro-Quer | C15H12O7 | 305.0650 | 289.631; 290.365 |
| 26 | 5.55 | [M+H]$^+$ | Flavonol base + 4O, 1MeO | C16H12O8 | 333.0602 | 58.065; 318.036 |
| 27 | 7.25 | [M+H]$^+$ | Naringenin-7-O-glucoside | C21H22O10 | 435.1279 | 153.0385; 273.0744 |
| 28 | 7.29 | [M+H]$^+$ | Phlorizin | C21H24O10 | 437.1445 | 107.045; 275.0905 |
| 29 | 7.75 | [M+H]$^+$ | Hyperoside | C21H20O12 | 465.1028 | 61.0285; 85.0285 |
| 30 | 8.22 | [M+H]$^+$ | Aurantio-obtusin beta-D-glucoside | C23H24O12 | 493.1329 | 331.0826 |
| 31 | 9.92 | [M+H]$^+$ | Kaempferol 3-glucorhamnoside | C27H30O15 | 595.1650 | 85.0305; 287.0686 |
| 32 | 9.92 | [M+H]$^+$ | Vicenin 2 | C27H30O15 | 595.1658 | 325.071; 317.0645 |

The EAA:NEAA ratio was 0.47, which does not meet the ideal protein condition proposed by the Food and Agriculture Organization of the United Nations and the World Health Organization [43]. Therefore, this fruit is not recommended as a high-quality protein food. The percentages of glutamic acid, glycine, and aspartic acid were 18.72%, 7.42%, and 9.54%, respectively, and accounting for >33% of the total amino acids in *B. heteropoda* fruit. Moreover, the UFA:SFA ratio in *B. heteropoda* fruit was 1.06, suggesting that it should not be recommended as a food rich in fatty acids.

Phenolic and flavonoid compounds are as important phytonutrients in plants [44, 45]. Flavonoids are secondary metabolites and abundant [46], and phenols are important plant compounds that mimic the biological effects of vitamin E [47]. A previous study reported that anthocyanins are rich in many plants and responsible for red, yellow, purple, black, and other colorful pigments [48]. The basic structural unit of anthocyanins is 2-phenylbenzopyran, which comprises a C6–C3–C6 backbone [49]. Its unique structure enables it to exert anti-

oxidation, anti-inflammation, and antitumor functions [50], as well as those related to the prevention of cardiovascular disease and enhancement of vision [51]. In the present study, we found that the anthocyanin content of *B. heteropoda* fruit was higher than that of wolfberry [39] but lower than that of *Passiflora foetida* [10]. Moreover, the TPC of *B. heteropoda* fruit was lower than barberry *(Berberis vulgaris L.), and* calafate *(Berberis microphylla)* fruits and other native berries, suggesting that the antioxidant activity of *B. heteropoda* is likely lower than that of several other *Berberis* fruits [52, 53]. A possible reason for this could be that the TPC is significantly related to geographical, climate, and soil conditions. Furthermore, we found the anthocyanin content in *B. heteropoda* fruit was inconsistent with that of a previous study [3], which reported a TAC of 20.37 mg/g fresh weight of *B. heteropoda*. A possible explanation could be that the samples in the present study were obtained from the Nanshan Mountain area of Urumqi City (latitude 89°29′36″E, longitude 43°27′32″N), whereas those in the previous study were from Daxigou (latitude 44°26′ N, longitude 80°46′ E).

A previous study reported the free radical-scavenging activity of flavonoids and polyphenols from *Stachys affinis* [54]. The $IC_{50}$ value is typically used to evaluate antioxidant activity, with smaller $IC_{50}$ values indicating stronger antioxidant capacity. The present results indicated that *B. heteropoda* fruit extract showed strong scavenging effects on DPPH•, •OH, $O_2^-$•, and $ABTS^+$•, suggesting that *B. heteropoda* fruit could be considered an excellent source of natural antioxidants.

This study has some limitations. First, we did not address the functional monomers of *B. heteropoda* fruits, nor did we assess the structures of specific phenolic compounds and their antioxidant effects. Furthermore, the mechanism of action for the phenolic compounds needs further evaluation, and the potential effect of *B. heteropoda* fruits on general human health needs further assessment.

## Conclusions

This study analyzed the major nutrients, mineral elements, fatty acids, and amino acids in *B. heteropoda* fruits and identified a wide array of important nutrient components. We found that *B. heteropoda* fruits had high TPC, TFC, and TAC values, as well as potentially excellent antioxidant properties. These findings suggest that *B. heteropoda* fruit could potentially be used as a health-promoting food for resisting oxidative damage; however, further studies are necessary to assess the biological activities of *B. heteropoda* fruit.

## Supporting information

**S1 Fig. Representative total ion chromatogram of an extract sample obtained from *Berberis heteropoda* fruit in positive ion mode (A) and negative ion mode (B) by UPLC-Q-TOF-MS**[E]**.**
(TIF)

**S1 File. Nutritional composition of *B. heteropoda* fruit.**
(XLSX)

**S2 File. Antioxidant Activity of *B. heteropoda* fruit extract.**
(XLSX)

**S3 File. Identification of Phenols in *B. heteropoda* fruit extract using chromatography and MS.**
(XLSX)

## Acknowledgments

The authors would like to thank expert Lude Xin for identifying the plants, Guangzhou Jinzhi Testing Technology Co., Ltd. for excellent technical assistance during this study, and Professor Yu Wei from Dalian Institute of Chemical Physics, Chinese Academy of Sciences, for editorial assistance with the manuscript.

## Author Contributions

**Conceptualization:** Jianguang Li.

**Data curation:** Jixiang Sun.

**Formal analysis:** Qian Li.

**Funding acquisition:** Jing Liu.

**Investigation:** Qian Li, Jing Liu.

**Methodology:** Jixiang Sun.

**Resources:** Jixiang Sun, Qian Li.

**Supervision:** Jianguang Li.

**Visualization:** Qian Li, Fang Xu.

**Writing – original draft:** Jixiang Sun, Fang Xu.

**Writing – review & editing:** Jianguang Li.

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
