## [Decision Letter · Decision Letter 0]

11 Nov 2021

PONE-D-21-29715Nutritional Composition and Antioxidant Properties of the Fruit of Berberis heteropoda SchrenkPLOS ONE

Dear Dr. Li,

Thank you for submitting your manuscript to PLOS ONE. After careful consideration, we feel that it has merit but does not fully meet PLOS ONE’s publication criteria as it currently stands. Therefore, we invite you to submit a revised version of the manuscript that addresses the points raised during the review process.

The authors have presented the whole manuscript in a rather indistinct fashion, gaps in the content are visible and the whole text lacks flow. Moreover, there are numerous places in the M&M section as well as in Results containing inconsistent methodology and vaguely presented results and conclusions. The manuscript would hugely benefit if being proofread by a senior researcher of the similar expertise, which would steer the authors how to properly present their experiments. It is also highly recommended to have the manuscript copyedited by a native English speaker or a professional editing agency towards gaining clarity and better readability. The authors are strongly encouraged to meticulously read and analyze Reviewers' reports and to complement or rectify the text where needed. The authors should also provide point-by-point reply to Reviewers' reports along with manuscript resubmission. Please be aware of additional comments of Reviewer #1 provided in the attachment.

We look forward to receiving your revised manuscript.

Kind regards,

Branislav T. Šiler, Ph.D.

Academic Editor

PLOS ONE

Journal Requirements:

Reviewers' comments:

Reviewer's Responses to Questions

**Comments to the Author**

1. Is the manuscript technically sound, and do the data support the conclusions?

Reviewer #1: Yes

Reviewer #2: No

Reviewer #3: Partly

2. Has the statistical analysis been performed appropriately and rigorously? 

Reviewer #1: Yes

Reviewer #2: No

Reviewer #3: Yes

3. Have the authors made all data underlying the findings in their manuscript fully available?

Reviewer #1: No

Reviewer #2: No

Reviewer #3: Yes

4. Is the manuscript presented in an intelligible fashion and written in standard English?

Reviewer #1: No

Reviewer #2: No

Reviewer #3: Yes

5. Review Comments to the Author

Reviewer #1: Manuscript No. PONE-D-21-29715

Manuscript Title: Nutritional Composition and Antioxidant Properties of the Fruit of Berberis heteropoda

Schrenk

Comments: The study discussed in this manuscript explains the major nutrients and antioxidant properties of Berberis heteropoda Schrenk fruits. Berberis plants are reported to possess antimicrobial, antiemetic, antipyretic, antioxidant, anti-inflammatory, anti-arrhythmic, sedative, anti-cholinergic, cholagogic, anti-leishmaniasis, and anti-malaria, and thus are important plants to provide bioactive secondary metabolites.

The present findings revealed that plant material under study is rich in phenolics which are considered as strong antioxidant, and can be part of nutraceuticals or functional foods.

In ABSTRACT, RESULTS: ……. in Berberis heteropoda Schrenk fruits were 75.22, 0.506, 2.55, 1.31, and 17.72 g100 g fresh fruit, probably it is 16.72g/100 g…?????

Similarly, …. The total phenol, flavonoid, and anthocyanin 34

contents of Berberis heteropoda Schrenk fruits were 68.55 mg gallic acid equivalentsg, 108.42 mg quercetin equivalentsg, and 19.83 mg cyanidin-3-glucoside equivalentg, should it be equivalent/g ?????

…The UPLC-Q-TOF-MSE analysis of phenols revealed 32 compounds……, what is mean by analysis of phenols???

Introduction: …..shrub of the family Berberidaceae and is distributed in… it should be …which is distributed in Altai, ……

Don’t use full name of the plant every time, for example …..and fruits of Berberis heteropoda Schrenk…. Can be written as and fruits of B. heteropoda…….

In introduction, language composition is not good, some parts are deleted, some are under lined please address this issue carefully. Several notes are not clear due to poor language

Introduction Line 66 states about ..recent study…, the reference quoted at 67 is of 2014, so it is not recent….

Provide voucher specimen No.????

Extraction: Line 125….. The samples were extracted using a modification method…extracted using reported method with modification….

Rewrite extraction part: it is again language problem

Determination of TPC: Line 136….Pure water means?? Distilled or deionized???

Nutritional Composition of the Berberis heteropoda Schrenk Fruit: Lines 237 and 238; Water content was the highest (75.22±1.75 g/100 g),…… Remove the word highest, just provide value.

Table 1: column composition per unit, ….remove repeated word “fresh fruit” and add in table caption.

Line 290 and onward… The details of 32 kinds of compounds are listed as follows….. Since all these compounds are listed in Table 4, it is just repetition in the text, so remove it, and better to add comments on percentage of components and their possible role in bioactivities.

Chromatograms and mass spectra of 32 compounds should be provided as supplementary material.

There are serious grammatical errors, language should be improved, repetition should be avoided

Reviewer #2: while going through the manuscript, the manuscript did not have much interesting finding and lacking novelty as far as my knowledge is concerned. The manuscript should include some biological activities like antimicrobial activity, anticancer etc.

Reviewer #3: PONE-D-21-29715. Nutritional Composition and Antioxidant Properties of the Fruit of Berberis heteropoda Schrenk

The subject of this manuscript falls within the general scope of the journal, and the study of Nutritional Composition and Antioxidant Properties of the Fruit of Berberis heteropoda Schrenk is relevant.

Keywords: the words Berberis heteropoda Schrenk; nutritional composition and antioxidant properties are in the title too. These keywords must be replaced by other ones.

Abstract: the precise units for the total content of phenol, flavonoid, and anthocyanin are not found this section, i.e. “…68.55 mg gallic acid equivalentsg…”. g of fresh fruit weight or dry fruit weight?

Introduction: this section contains the information that justifies this work. The antecedents on B. heteropoda Schrenk fruits are described, although briefly. The objective of the manuscript is correctly stated.

Line 47: “…The roots, bark, stems, and fruits of Berberis heteropoda Schrenk…”. Delete the term Schrenk in this sentence and in the following ones.

Materials and Methods: even though this section includes details about the methods employed, several topics remain unclear.

Line 74: “…Mature Berberis heteropoda Schrenk fruits were collected…”. The correct term is ripe rather than mature.

Line 76: “…Fig. 1 shows its distribution in Xinjiang…”. Show the site where the fruits were collected. Site 1, 2 or 3 or all? This Figure must show the geographical position.

Line 79: “…High-quality Berberis heteropoda Schrenk fruit were selected…” The term “High quality” must be more precise. Nothing is said about the sample size and the number of shrubs selected for the fruit harvest.

Line 125-132. The different solvents used for the fruit extraction is not described in this section, i.e. methanol, acetone and ethanol. It is necessary to add this information here.

Line 140: “…TPC was denoted as milligram of gallic acid equivalent per gram of plant mass…”. Plant mass must be replaced by fruit mass. Fresh or dry fruit weight?

Line 149-150: The same comment for Line 140.

Results: this section is presented in 2 Figures and 3 Tables. Legends of Figures 2 and 3 do not content sufficient information about the statistical analysis of the results, i.e. the meaning of the asterisks above the bars.

Line 266-267: “…The regression equations used were y = 0.0109x + 0.0157 and y = 0.067x - 0.0173…”. Rewrite as “…The regression equations used were y = 0.0109x + 0.0157 and y = 0.067x - 0.0173, respectively…”.

Line 270: “…the TFC, TPC, and TAC values were 108.42, 68.55, and 19.83 mg/g fruit, 271 respectively…” Please add the term of expression referred, i.e. mg/g fresh fruit weight.

Discussion:

Lines 317-318: this sentence is more suitable for the Introduction section.

Lines 327-328: the term “this study” means the study of reference 3 or the study of this manuscript? Please, be clearer.

Lines 362-364: “…The anthocyanin content of Berberis heteropoda Schrenk fruit found in this study was slightly inconsistent with a prior study, possibly because the previous samples were obtained from Yili, Xinjiang [3]…”. The meaning of this sentence must be clearer.

As a general comment of the Discussion section, the authors do not discuss the obtained results with other Berberis fruits species, in particular with respect to TPC, TFC and TAC.

Conclusions: the conclusions are well stated.

References: the inclusion of references of other Berberis fruit species is suggested to be discussed.

Final comment: this manuscript needs to incorporate the corrections suggested before its publication in PLOS ONE.

6. PLOS authors have the option to publish the peer review history of their article (what does this mean?). If published, this will include your full peer review and any attached files.

Reviewer #1: No

Reviewer #2: No

Reviewer #3: No

---

## [Author Response · Author response to Decision Letter 0]

6 Dec 2021

Point-By-Point Response

Journal Requirements:

Question 1: Please ensure that your manuscript meets PLOS ONE's style requirements, including those for file naming. The PLOS ONE style templates can be found at 

Response: Thanks for this suggestion, and the style of manuscript have already adjusted according to PLOS ONE style template. 

Question 2: PLOS requires an ORCID iD for the corresponding author in Editorial Manager on papers submitted after December 6th, 2016. Please ensure that you have an ORCID iD and that it is validated in Editorial Manager. To do this, go to ‘Update my Information’ (in the upper left-hand corner of the main menu), and click on the Fetch/Validate link next to the ORCID field. This will take you to the ORCID site and allow you to create a new iD or authenticate a pre-existing iD in Editorial Manager. Please see the following video for instructions on linking an ORCID iD to your Editorial Manager account: https://www.youtube.com/watch?v=_xcclfuvtxQ

Response: Thanks for this suggestion, and the ORCID iD for the corresponding author have already added in the revised manuscript.

Question 3: We note that Figure 1 in your submission contain [map/satellite] images which may be copyrighted. All PLOS content is published under the Creative Commons Attribution License (CC BY 4.0), which means that the manuscript, images, and Supporting Information files will be freely available online, and any third party is permitted to access, download, copy, distribute, and use these materials in any way, even commercially, with proper attribution. For these reasons, we cannot publish previously copyrighted maps or satellite images created using proprietary data, such as Google software (Google Maps, Street View, and Earth). For more information, see our copyright guidelines: http://journals.plos.org/plosone/s/licenses-and-copyright.

Response: Thanks for this suggestion, and the Figure 1 have already removed in the revised manuscript. 

Response to reviewer #1

General comments: The study discussed in this manuscript explains the major nutrients and antioxidant properties of Berberis heteropoda Schrenk fruits. Berberis plants are reported to possess antimicrobial, antiemetic, antipyretic, antioxidant, anti-inflammatory, anti-arrhythmic, sedative, anti-cholinergic, cholagogic, anti-leishmaniasis, and anti-malaria, and thus are important plants to provide bioactive secondary metabolites.

The present findings revealed that plant material under study is rich in phenolics which are considered as strong antioxidant, and can be part of nutraceuticals or functional foods.

Response: As behalf of all co-authors, I would like to appreciate this referee due to thoughtful comments proposed by the peer review. After we revised the manuscript, those significant issues could be changed.

Question 1: In ABSTRACT, RESULTS: ……. in Berberis heteropoda Schrenk fruits were 75.22, 0.506, 2.55, 1.31, and 17.72 g100 g fresh fruit, probably it is 16.72g/100 g…?????Similarly, …. The total phenol, flavonoid, and anthocyanin 34 contents of Berberis heteropoda Schrenk fruits were 68.55 mg gallic acid equivalentsg, 108.42 mg quercetin equivalentsg, and 19.83 mg cyanidin-3-glucoside equivalentg, should it be equivalent/g ?????

Response: Thanks for this suggestion, and these sentences have already changed in the revised manuscript.

Question 2: The UPLC-Q-TOF-MSE analysis of phenols revealed 32 compounds……, what is mean by analysis of phenols???

Response: Thanks for this suggestion, and this sentence have already changed into: “The UPLC-Q-TOF-MSE analysis of polyphenols of Berberis heteropoda Schrenk fruit revealed 32 compounds”

Question 3: …Introduction: …..shrub of the family Berberidaceae and is distributed in… it should be …which is distributed in Altai, ……

Response: Thanks for this suggestion, and the sentence have already changed in the revised manuscript.

Question 4: Don’t use full name of the plant every time, for example …..and fruits of Berberis heteropoda Schrenk…. Can be written as and fruits of B. heteropoda…….

Response: Thanks for this suggestion, and the name of the plant have already changed in the revised manuscript. 

Question 5: In introduction, language composition is not good, some parts are deleted, some are under lined please address this issue carefully. Several notes are not clear due to poor language

Response: Thanks for this suggestion, and the English revision have already performed in the revised manuscript by Editage Company. 

Question 6: Introduction Line 66 states about ..recent study…, the reference quoted at 67 is of 2014, so it is not recent….

Response: Thanks for this suggestion, and this sentence have already changed in the revised manuscript. 

Question 7: Provide voucher specimen No.????

Response: Thanks for this suggestion, and voucher specimen No have already added in the revised manuscript. 

Question 8: Extraction: Line 125….. The samples were extracted using a modification method…extracted using reported method with modification….Rewrite extraction part: it is again language problem

Response: Thanks for this suggestion, and this sentence have already changed in the revised manuscript. Moreover, the language of whole manuscript have already revised by Editage Company. 

Question 9: Determination of TPC: Line 136….Pure water means?? Distilled or deionized???

Response: Thanks for this suggestion, and pure water have already changed into distilled water.

Question 10: Nutritional Composition of the Berberis heteropoda Schrenk Fruit: Lines 237 and 238; Water content was the highest (75.22±1.75 g/100 g),…… Remove the word highest, just provide value.

Response: Thanks for this suggestion, and this sentence have already changed into: “Water content was 75.22±1.75 g/100 g, crude fiber was 17.72±0.52 g/100 g and total protein was 2.55±0.03 g/100 g.”

Question 11: Table 1: column composition per unit, ….remove repeated word “fresh fruit” and add in table caption.

Response: Thanks for this suggestion, and this change have already performed in the revised Table 1. 

Question 12: Line 290 and onward… The details of 32 kinds of compounds are listed as follows….. Since all these compounds are listed in Table 4, it is just repetition in the text, so remove it, and better to add comments on percentage of components and their possible role in bioactivities.

Response: Thanks for this suggestion, and these sentences have already removed in the revised manuscript. 

Question 13: Chromatograms and mass spectra of 32 compounds should be provided as supplementary material.

Response: Thanks for this suggestion, and the Chromatograms and mass spectra of 32 compounds have already removed in Figure S1.

Question 14: There are serious grammatical errors, language should be improved, repetition should be avoided

Response: Thanks for this suggestion, and English revision have already performed by Editage Company.

Response to reviewer #2

General comments: while going through the manuscript, the manuscript did not have much interesting finding and lacking novelty as far as my knowledge is concerned. The manuscript should include some biological activities like antimicrobial activity, anticancer etc.

Response: Thanks for this suggestion. The current study aimed to assess the major nutrients and antioxidant properties of B. heteropoda fruits, while the biological activities were not illustrated. We have already addressed this comment in Discussion section.

Response to reviewer #3

General comments: The subject of this manuscript falls within the general scope of the journal, and the study of Nutritional Composition and Antioxidant Properties of the Fruit of Berberis heteropoda Schrenk is relevant.

Response: As behalf of all co-authors, I would like to appreciate this referee due to thoughtful comments proposed by the peer review. After we revised the manuscript, those significant issues could be changed.

Question 1: Keywords: the words Berberis heteropoda Schrenk; nutritional composition and antioxidant properties are in the title too. These keywords must be replaced by other ones.

Response: Thanks for this suggestion, and the keywords have already changed into: “Amino acids; minerals; fatty acids; total phenol; flavonoid; anthocyanin”.

Question 2: Abstract: the precise units for the total content of phenol, flavonoid, and anthocyanin are not found this section, i.e. “…68.55 mg gallic acid equivalentsg…”. g of fresh fruit weight or dry fruit weight?

Response: Thanks for this suggestion, and the results in Abstract section have already changed into: “The analytical results showed that average water, total fat, total protein, ash, and carbohydrates contents in B. heteropoda fruits were 75.22 g/100 g, 0.506 g/100 g, 2.55 g/100 g, 1.31 g/100 g, and 17.72 g/100 g fresh fruit, respectively. The total phenol, flavonoid, and anthocyanin contents of B. heteropoda fruits were 68.55 mg gallic acid equivalents/g, 108.42 mg quercetin equivalents/g, and 19.83 mg cyanidin-3-glucoside equivalent/g, respectively. The UPLC-Q-TOF-MSE analysis of polyphenols of B. heteropoda fruit revealed 32 compounds. ”

Question 3: Introduction: this section contains the information that justifies this work. The antecedents on B. heteropoda Schrenk fruits are described, although briefly. The objective of the manuscript is correctly stated.

Response: We appreciate the reviewer given this kindly comment.

Question 4: Line 47: “…The roots, bark, stems, and fruits of Berberis heteropoda Schrenk…”. Delete the term Schrenk in this sentence and in the following ones.

Response: Thanks for this suggestion, and these changes have already performed in the revised manuscript.

Question 5: Materials and Methods: even though this section includes details about the methods employed, several topics remain unclear.

Response: As behalf of all co-authors, I would like to appreciate this referee due to thoughtful comments proposed by the peer review. After we revised the manuscript, those significant issues could be changed.

Question 6: Line 74: “…Mature Berberis heteropoda Schrenk fruits were collected…”. The correct term is ripe rather than mature.

Response: Thanks for this suggestion, and this term have already changed into: “ripe”.

Question 7: Line 76: “…Fig. 1 shows its distribution in Xinjiang…”. Show the site where the fruits were collected. Site 1, 2 or 3 or all? This Figure must show the geographical position.

Response: Thanks for this suggestion, and the Figure 1 have already removed in the revised manuscript, and the details of geographical position have already described.

Question 8: Line 79: “…High-quality Berberis heteropoda Schrenk fruit were selected…” The term “High quality” must be more precise. Nothing is said about the sample size and the number of shrubs selected for the fruit harvest.

Response: Thanks for this suggestion, and the following sentences have already added in the revised manuscript: “B. heteropoda fruit were selected after combed, and all of fruit were evenly and full, without rotten and miscellaneous, then the stem and seeds were removed. A total of 3 kg B. heteropoda fruit were collected from dozens of shrubs in a ravine.”

Question 9: Line 125-132. The different solvents used for the fruit extraction is not described in this section, i.e. methanol, acetone and ethanol. It is necessary to add this information here.

Response: Thanks for this suggestion, and this information have already added in the revised manuscript.

Question 10: Line 140: “…TPC was denoted as milligram of gallic acid equivalent per gram of plant mass…”. Plant mass must be replaced by fruit mass. Fresh or dry fruit weight?

Response: Thanks for this suggestion. We have already made this change in the revised manuscript. 

Question 11: Line 149-150: The same comment for Line 140.

Response: Thanks for this suggestion. We have already made this change in the revised manuscript. 

Question 12: Results: this section is presented in 2 Figures and 3 Tables. Legends of Figures 2 and 3 do not content sufficient information about the statistical analysis of the results, i.e. the meaning of the asterisks above the bars.

Response: Thanks for this suggestion, and the meaning of the asterisks have already added in Figures legends. 

Question 13: Line 266-267: “…The regression equations used were y = 0.0109x + 0.0157 and y = 0.067x - 0.0173…”. Rewrite as “…The regression equations used were y = 0.0109x + 0.0157 and y = 0.067x - 0.0173, respectively…”.

Response: Thanks for this suggestion, and this changes have already performed in the revised manuscript. 

Question 14: Line 270: “…the TFC, TPC, and TAC values were 108.42, 68.55, and 19.83 mg/g fruit, 271 respectively…” Please add the term of expression referred, i.e. mg/g fresh fruit weight.

Response: Thanks for this suggestion, and this change have already performed in the revised manuscript. 

Question 15: Discussion:

Lines 317-318: this sentence is more suitable for the Introduction section.

Response: Thanks for this suggestion, and this sentence have already removed in the revised manuscript. 

Question 16: Lines 327-328: the term “this study” means the study of reference 3 or the study of this manuscript? Please, be clearer.

Response: Thanks for this suggestion, and “this study” have already changed into: “our study”. 

Question 17: Lines 362-364: “…The anthocyanin content of Berberis heteropoda Schrenk fruit found in this study was slightly inconsistent with a prior study, possibly because the previous samples were obtained from Yili, Xinjiang [3]…”. The meaning of this sentence must be clearer.

Response: Thanks for this suggestion, and this sentence have already revised in the revised manuscript.

Question 18: As a general comment of the Discussion section, the authors do not discuss the obtained results with other Berberis fruits species, in particular with respect to TPC, TFC and TAC.

Response: Thanks for this suggestion, and several sentences have already added in the revised manuscript. 

Question 19: Conclusions: the conclusions are well stated.

Response: We appreciate the reviewer given this kindly comments.

Question 20: References: the inclusion of references of other Berberis fruit species is suggested to be discussed.

Response: Thanks for this suggestion, and the inclusion of references have already discussed in the revised manuscript.

Question 21: Final comment: this manuscript needs to incorporate the corrections suggested before its publication in PLOS ONE.

Response: We appreciate the reviewer given this kindly comments, and all of changes have already performed in the revised manuscript with tracked.

---

## [Decision Letter · Decision Letter 1]

15 Dec 2021

PONE-D-21-29715R1Nutritional Composition and Antioxidant Properties of the Fruit of Berberis heteropoda SchrenkPLOS ONE

Dear Dr. Li,

Thank you for submitting your manuscript to PLOS ONE. After careful consideration, we feel that it has merit but does not fully meet PLOS ONE’s publication criteria as it currently stands. Therefore, we invite you to submit a revised version of the manuscript that addresses the points raised during the review process.

The authors need to thoroughly reorganize the Discussion section, since some Reviewers' concerns were not properly addressed. Up-to-date literature that deals with the same topic, basically with bioactivity of other *Berberis *taxa must be consulted. Moreover, the authors probably missed to see the Academic Editor's comments in the previous round, since they were left unaddressed as well. I will repeat them:

"The authors have presented the whole manuscript in a rather indistinct fashion, gaps in the content are visible and the whole text lacks flow. Moreover, there are numerous places in the M&M section as well as in Results containing inconsistent methodology and vaguely presented results and conclusions. The manuscript would hugely benefit if being proofread by a senior researcher of the similar expertise, which would steer the authors how to properly present their experiments. It is also highly recommended to have the manuscript copyedited by a native English speaker or a professional editing agency towards gaining clarity and better readability."

Therefore, language polishing is much needed throughout the text.

In L27 please do not introduce the abbreviated plant species name in parenthesis. It is a common scientific practice to abbreviate the genus name once being introduced in full.

We look forward to receiving your revised manuscript.

Kind regards,

Branislav T. Šiler, Ph.D.

Academic Editor

PLOS ONE

Additional Editor Comments (if provided):

The authors need to thoroughly reorganize the Discussion section, since some Reviewers' concerns were not properly addressed. Up-to-date literature that deals with the same topic, basically with bioactivity of other *Berberis *taxa must be consulted. Moreover, the authors probably missed to see the Academic Editor's comments in the previous round, since they were left unaddressed as well. I will repeat them:

"The authors have presented the whole manuscript in a rather indistinct fashion, gaps in the content are visible and the whole text lacks flow. Moreover, there are numerous places in the M&M section as well as in Results containing inconsistent methodology and vaguely presented results and conclusions. The manuscript would hugely benefit if being proofread by a senior researcher of the similar expertise, which would steer the authors how to properly present their experiments. It is also highly recommended to have the manuscript copyedited by a native English speaker or a professional editing agency towards gaining clarity and better readability."

Therefore, language polishing is much needed throughout the text. In L27 please do not introduce the abbreviated plant species name in parenthesis. It is a common scientific practice to abbreviate the genus name once being introduced in full.

Reviewers' comments:

Reviewer's Responses to Questions

**Comments to the Author**

1. If the authors have adequately addressed your comments raised in a previous round of review and you feel that this manuscript is now acceptable for publication, you may indicate that here to bypass the “Comments to the Author” section, enter your conflict of interest statement in the “Confidential to Editor” section, and submit your "Accept" recommendation.

Reviewer #1: All comments have been addressed

Reviewer #3: (No Response)

2. Is the manuscript technically sound, and do the data support the conclusions?

Reviewer #1: Partly

Reviewer #3: Yes

3. Has the statistical analysis been performed appropriately and rigorously? 

Reviewer #1: Yes

Reviewer #3: Yes

4. Have the authors made all data underlying the findings in their manuscript fully available?

Reviewer #1: Yes

Reviewer #3: Yes

5. Is the manuscript presented in an intelligible fashion and written in standard English?

Reviewer #1: No

Reviewer #3: Yes

6. Review Comments to the Author

Reviewer #1: nothing special, ,most of the corrections are addressed but language still needs serious revision. In my first review, I strongly suggested to edit language by a native English speaker.

Reviewer #3: PONE-D-21-29715R1. Nutritional Composition and Antioxidant Properties of the Fruit of Berberis heteropoda Schrenk

The authors have incorporated most of the suggestions made by the reviewers. However the following items must be improved:

Discussion:

Lines 344-346: “…Our study found the anthocyanin content in B. heteropoda fruit was inconsistent with a prior study [3], which indicated the TAC was 20.37 mg/g fresh weight of B. heteropoda...".Are the differences between 19.83 and 20.37 mg cyanidin-3-glucoside equivalent/g fresh weight significant?

As a general comment of the Discussion section, the authors do not discuss the obtained results with other Berberis fruits species, in particular with respect to TPC, TFC and TAC. New bibliography on other Berberis species were not included in the revised manuscript.

Final comment: this manuscript needs to incorporate the corrections suggested before its publication in PLOS ONE.

7. PLOS authors have the option to publish the peer review history of their article (what does this mean?). If published, this will include your full peer review and any attached files.

Reviewer #1: No

Reviewer #3: No

---

## [Author Response · Author response to Decision Letter 1]

28 Dec 2021

Point-By-Point Response

Response to editor

General comments: The authors need to thoroughly reorganize the Discussion section, since some Reviewers' concerns were not properly addressed. Up-to-date literature that deals with the same topic, basically with bioactivity of other Berberis taxa must be consulted. Moreover, the authors probably missed to see the Academic Editor's comments in the previous round, since they were left unaddressed as well. I will repeat them:

"The authors have presented the whole manuscript in a rather indistinct fashion, gaps in the content are visible and the whole text lacks flow. Moreover, there are numerous places in the M&M section as well as in Results containing inconsistent methodology and vaguely presented results and conclusions. The manuscript would hugely benefit if being proofread by a senior researcher of the similar expertise, which would steer the authors how to properly present their experiments. It is also highly recommended to have the manuscript copyedited by a native English speaker or a professional editing agency towards gaining clarity and better readability."

Therefore, language polishing is much needed throughout the text. In L27 please do not introduce the abbreviated plant species name in parenthesis. It is a common scientific practice to abbreviate the genus name once being introduced in full.

Response: As behalf of all co-authors, I would like to appreciate this referee due to thoughtful comments proposed by the peer review. After we revised the manuscript, those significant issues could be changed. 

As compared with other Berberis taxa have already added in Discussion. Moreover, the methods and results section were adjusted to address your comment. Furthermore, the English revision in the revised manuscript have already performed by Editage Company. Finally, the abbreviated plant species name in parenthesis have already removed in the revised manuscript. 

Response to reviewer #1

General comments: nothing special, ,most of the corrections are addressed but language still needs serious revision. In my first review, I strongly suggested to edit language by a native English speaker.

Response: Thanks for this suggestion, and the English revision is provided by the Editage Company.

Response to reviewer #3

General comments: The authors have incorporated most of the suggestions made by the reviewers. However the following items must be improved:

Response: As behalf of all co-authors, I would like to appreciate this referee due to thoughtful comments proposed by the peer review. After we revised the manuscript, those significant issues could be changed. 

Question 1: Lines 344-346: “…Our study found the anthocyanin content in B. heteropoda fruit was inconsistent with a prior study [3], which indicated the TAC was 20.37 mg/g fresh weight of B. heteropoda...”. Are the differences between 19.83 and 20.37 mg cyanidin-3-glucoside equivalent/g fresh weight significant?

Response: Thanks for this suggestion. According to the standard deviation of TAC in our study and prior study, we noted the difference was associated with statistically significant. 

Question 2: As a general comment of the Discussion section, the authors do not discuss the obtained results with other Berberis fruits species, in particular with respect to TPC, TFC and TAC. New bibliography on other Berberis species were not included in the revised manuscript.

Response: Thanks for this suggestion, and several sentences have already added in Discussion section to compare the obtained results with other Berberis fruits.

Question 3: Final comment: this manuscript needs to incorporate the corrections suggested before its publication in PLOS ONE.

Response: Thanks for this suggestion, and all of changes have already performed to address your comments.

---

## [Editor Report · Decision Letter 2]

31 Dec 2021

Nutritional Composition and Antioxidant Properties of the  Fruit of Berberis heteropoda Schrenk

PONE-D-21-29715R2

Dear Dr. Li,

We’re pleased to inform you that your manuscript has been judged scientifically suitable for publication and will be formally accepted for publication once it meets all outstanding technical requirements.

Kind regards,

Branislav T. Šiler, Ph.D.

Academic Editor

PLOS ONE
---

## [Editor Report · Acceptance letter]

29 Mar 2022

PONE-D-21-29715R2 

Nutritional Composition and Antioxidant Properties of the Fruit of *Berberis heteropoda* Schrenk 

Dear Dr. Li:

I'm pleased to inform you that your manuscript has been deemed suitable for publication in PLOS ONE. Congratulations! Your manuscript is now with our production department. 

Kind regards, 

on behalf of

Dr. Branislav T. Šiler 

Academic Editor

PLOS ONE